# The Influence of Machining Conditions on the Orientation of Nanocrystallites and Anisotropy of Physical and Mechanical Properties of Flexible Graphite Foils

**DOI:** 10.3390/nano14060540

**Published:** 2024-03-19

**Authors:** Vladimir A. Shulyak, Nikolai S. Morozov, Andrei V. Ivanov, Alexandra V. Gracheva, Sergei N. Chebotarev, Viktor V. Avdeev

**Affiliations:** Department of Chemistry, Lomonosov Moscow State University, 119234 Moscow, Russia; key700@mail.ru (A.V.I.); gracheva.a@inumit.ru (A.V.G.); chebotarev.s@inumit.ru (S.N.C.); avdeev@highp.chem.msu.ru (V.V.A.)

**Keywords:** flexible graphite foil, natural graphite, thermally expanded graphite, rolling, X-ray diffraction, SEM, TEM, anisotropy, tensile strength, elastic characteristics, coherent scattering regions, misorientation angles

## Abstract

The physical and mechanical properties and structural condition of flexible graphite foils produced by processing natural graphite with nitric acid, hydrolysis, thermal expansion of graphite and subsequent rolling were studied. The processes of obtaining materials and changing their characteristics has been thoroughly described and demonstrated. The structural transformations of graphite in the manufacture of foils were studied by X-ray diffraction analysis (XRD) and transmission electron microscopy (TEM). A decrease in the average size of the coherent scattering regions (CSR) of nanocrystallites was revealed during the transition from natural graphite to thermally expanded graphite from 57.3 nm to 20.5 nm at a temperature of 900 °C. The rolling pressure ranged from 0.05 MPa to 72.5 MPa. The thickness of the flexible graphite foils varied from 0.11 mm to 0.75 mm, the density—from 0.70 to 1.75 g/cm^3^. It was shown that with an increase in density within these limits, the compressibility of the graphite foil decreased from 65% to 9%, the recoverability increased from 5% to 60%, and the resiliency decreased from 10% to 6%, which is explained by the structural features of nanocrystallites. The properties’ anisotropy of graphite foils was studied. The tensile strength increased with increasing density from 3.0 MPa (ρ = 0.7 g/cm^3^) to 14.0 MPa (ρ = 1.75 g/cm^3^) both in the rolling direction L and across T. At the same time, the anisotropy of physical and mechanical properties increased with an increase in density along L and T to 12% with absolute values of 14.0 MPa against 12.5 MPa at a thickness of 200 μm. Expressed anisotropy was observed along L and T when studying the misorientation angles of nanocrystallites: at ρ = 0.7 g/cm^3^, it was from 13.4° to 14.4° (up to 5% at the same thickness); at ρ = 1.3 g/cm^3^—from 11.0° to 12.8° (up to 7%); at ρ = 1.75 g/cm^3^—from 10.9° to 12.4° (up to 11%). It was found that in graphite foils, there was an increase in the coherent scattering regions in nanocrystallites with an increase in density from 24.8 nm to 49.6 nm. The observed effect can be explained by the coagulation of nanocrystallites by enhancing the Van der Waals interaction between the surface planes of coaxial nanocrystallites, which is accompanied by an increase in microstrains. The results obtained can help discover the mechanism of deformation of porous graphite foils. The obtained results can help discover the deformation mechanism of porous graphite foils. We assume that this will help predict the material behavior under industrial operating conditions of products based flexible graphite foils.

## 1. Introduction

The natural graphite (NG) is used in numerous fabrications of various materials: melting crucibles, lining plates used in metal casting; electrodes and heating elements installed in corrosive media; lubricants; plastic fillers; neutron moderator rods in nuclear reactors; pencil rods, etc. [1,2,3,4,5,6,7,8,9,10]. Another notable application of natural graphite is in the manufacturing of graphite foils (GFs), which are utilized in sealing products for flange connections across several industries, including thermal and nuclear energy, oil and gas production, automotive, and chemical current sources for heat dissipation [11,12,13,14,15,16,17,18,19,20].

This paper explores the physical, mechanical properties [19,20,21,22] and GF’s structural characteristics [23] depending on the rolling modes under identical conditions for obtaining material from natural graphite for their manufacture. The process of producing GFs involves several steps. Initially, purified natural flake graphite undergoes treatment with strong Brønsted acids in the presence of an oxidizing agent to produce a graphite intercalation compound (GIC) [24,25]. Subsequently, the GIC is treated with water to obtain oxidized graphite (OG) [26,27] followed by heat treatment of OG to yield a powder of thermally expanded graphite (TEG) [11,17,28,29,30]. Notably, TEG possesses the unique ability to be compacted into dense forms without a binder, leading to the compression of TEG into flexible GFs [12,13,14,15,19].

Various methods exist for obtaining GIC through the direct interaction of graphite with acids (HNO_3_, H_2_SO_4_, HClO_4_, etc.) and oxidants (K_2_Cr_2_O_7_, KMnO_4_, H_2_O_2_, etc.) if necessary [24,25,26,27]. Studies indicate [12,15,31,32] that the properties of the initial graphite, the nature, and the method of obtaining GIC and TEG significantly impact the degree of TEG expansion and the properties of the resulting GFs [33]. When pressing the TEG, the TEG particles first mechanically adhere to each other, then they converge and increase the contacts between the graphite packs that make up the TEG particles. With an increase in density, the number of contacts between graphite packs increases, which leads to a regular decrease in the porosity of the material and better adhesion of particles to each other [34,35]. This leads to an increase in the electrical and heat-conducting properties of the material [36,37,38,39,40], while its gas permeability decreases [19,27,41,42].

The GFs ability to be pressed and restored formed the basis for the sealing products manufacturing of various types: from sealing rings to packing glands [43,44]. Additionally, the high temperature resistance and chemical inertness of graphite expand the range of sealed assemblies, machines and aggregates, increase the parameters of the working environment and simultaneously reduce the proportion of leaks compared to classical rubber and asbestos-containing materials [45]. The high compressibility of GFs allows for clamping rings between two sealed surfaces, under certain loads (up to 250 MPa), while its recoverability enables it to fill irregularities of the sealed surface and ensure long term tightness (up to 80 years) even after relaxation and load reduction on the seal, which occurs during long-term operation [25].

The amount of load applied during the operation of GF’s products depends on how much the material will be compacted and how much it will recover to fill the flange irregularities. At the same time, the ultimate operating parameters of the flange connection depend on the strength of the material. These parameters are significantly influenced by the value of the density and thickness of the GFs, as well as the structural state of the foil material, which in turn depends on the mechanical processing of the raw material from which it is obtained [46].

The objective of our study was to discover structural transformations in the nanocrystals by flexible graphite foils that occur during the material production process, and to investigate the influence of this changes on the material mechanical properties.

The processes of material preparation (natural graphite−graphite intercalated compound−oxidized graphite−thermally expanded graphite−flexible graphite foil) and their structural changes were characterized by SEM and X-ray diffraction (XRD). The fractional composition of pure natural graphite and its ash was calculated. The structural transformations in GFs were described by SEM, TEM and XRD. Also, the coherent scattering regions of GF’s nanocrystallites were calculated. The misorientation angles of the nanocrystallites were calculated by the XRD-method of recording the rocking curves. The physical and mechanical properties of the GFs were investigated by tensile test along and across the rolling direction. Their elastic characteristics were also studied.

## 2. Materials and Test Methods

### 2.1. Production of Flexible Graphite Foils

The production of GFs took place in several stages. In this study, natural purified flake graphite with an ash content of 3.4 ± 0.4 wt% produced by INGRAF was used. The fractional composition is shown in Figure 1.

Then, natural graphite was subjected to additional purification in the laboratory, to an ash content of ~0.3 wt%. To obtain all the samples studied in this work, laboratory-purified graphite (PG) was used. The fractional composition of PG was studied by the Cilas 1180 LD laser fractional analyzer in a surfactant solution. The material was placed in a solution, sonicated for 5 seconds, and then irradiated with a laser while stirring.

In the second stage of GF’s production, additionally purified natural graphite was intercalated with highly concentrated nitric acid (98%) [47]. Graphite and acid were mixed in a ratio of 1 to 0.8 by weight, and a GIC was obtained (Figure 2a). This mixture was stirred for 1 h. The mixture was then washed with distilled water at a ratio of 1 to 30. Thus, OG was produced (Figure 2b), which was then separated from the water on a glass porous filter and dried for 5 h in an oven at a temperature of 50 °C.

In the third stage, the OG was heat treated at 900 °C for 5 s. This process involved passing it through a pipe inserted into the furnace in the air flow. As a result, the flakes of the exhaust gas were opened and TEG was obtained (Figure 2c).

In the fourth stage, there was primary rolling of TEG into GF’s sheets (Figure 2d) with thicknesses of 0.75; 0.5; 0.28 mm and density of 0.70 g/cm^3^. Detailed parameters are presented in Table 1, and the GFs obtained by primary rolling is highlighted in green. The GF’s sheets obtained during primary machining were subjected to additional rolling to densities of 1.0, 1.3, 1.6, and 1.75 g/cm^3^ with corresponding thickness reductions. The total number of samples tested in the study was more than 400 units. The GFs parameters were determined for 20 samples of each position in the Table 1.

### 2.2. Measurement of Flexible Graphite Foil Density

Density measurements were performed as follows. Using a caliper (deviation ≤ 0.01 mm), the length and width of the GF sheet were measured several times, the value was then averaged. A micrometer (deviation ≤ 4 μm) was used to measure the thickness in three places, and then the value was averaged. The weight of the samples was measured using weights (deviation ≤ 0.001 g). In the extreme case, the density measurement deviation was less than 1%, and this maximum value was extended to all measurements.

### 2.3. Study of the Fractional Composition of Graphite Ash

Determination of ash content and fractional composition of GF ash was carried out according to ASTM C561-23 [48]. To do this, GF weighing 5 g was stirred in a pre-calcinated porcelain crucible, weighed on the Shimadzu AX200 scale (Kyoto, Japan), and kept in a muffle furnace for 24 h at 1000 °C.

The crucible was placed in a desiccator and cooled to room temperature. Then it was weighed with the ash. Next, the crucible with ash was put back into the furnace, and the procedure was repeated until the mass of the crucible with ash stopped changing. These values were taken as final, and the ash content was calculated by (1):x = (m_c+ash_ − m_c_)/(m_c+gf_ − m_c_)·100%,(1)
where (m_c_) is the mass of the crucible, (m_c+gf_) is the mass of the crucible and GF, (m_c+ash_) is the mass of the crucible and ash, and x is the ash content of GF in percent.

### 2.4. Examination of Foil Samples Using SEM

The morphology and structure of PG, OG, TEG flakes and GF were studied on a TESCAN VEGA 3 scanning electron microscope (SEM) (Brno, Czech Republic) by the lanthanum hexaboride cathode and the secondary electron detector SE. An accelerating voltage of HV = 10 kV was used to obtain all SEM images.

Also, using the image processing tools of the device, the dimensions of the PG, OG, TEG flakes and GFs were calculated. The adherence pattern of the graphite flakes to each other in the foil was shown.

### 2.5. The GFs Samples Investigation by TEM

A TITAN™ transmission electron microscope (FEI, Hilsboro, OR, USA) was used to study the structure and nature of nanocrystallite stacking in GF. Lamellas preparation was carried out on SEM FEI SCIOS 2 (Figure 3).

From the TEM images, the convergence of graphite flakes was assessed by the values of the angles between them.

### 2.6. XRD Analysis of the Test Samples

X-ray phase and structural analysis of PG, GIC, OG, TEG, and GF were carried out on Rigaku Ultima IV diffractometer (Tokyo, Japan). The study of the samples was carried out with Bragg-Brentano focusing, CuKα radiation = 0.15425 nm, in increments of 0.02°, at a rate of 2°/min in the *θ*/2*θ* diffractometer mode [49].

The reflex of the diffraction pattern is a convolution of several contributions of different components [50,51]:(2)$$FWHM=\beta_{\mathrm{physical}}+\beta_{\mathrm{instr}}+\beta_{\mathrm{stress}},$$
where FWHM is the value of the full width at half maximum of the reflex of the experimental diffractogram; $\beta_{\mathrm{physical}}$ is the value of the physical broadening of the X-ray reflection caused by the fine dispersion of the nanocrystallites of the sample; $\beta_{\mathrm{instr}}$ is the value of the instrumental broadening of the X-ray reflection at its width at half maximum caused by the diffractometer geometry and characteristics; $\beta_{\mathrm{stress}}$ is the value of the broadening at the width at half maximum of the reflex arising from microstrains.

The structural state of the material was assessed by the values of the interplanar distance *d*, $\beta_{\mathrm{physical}}$—the physical broadening of the reflex, the parameter of the graphite crystal lattice *c* in GF and the coherent scattering regions (CSR) D.

The calculation of the size of the CSR $D_{\mathrm{Sch}}$ was carried out by two methods, which were then compared with each other. In the first case, the calculation was made using the method proposed by Scherrer [52,53]:(3)$$\beta_{\mathrm{physical}}=FWHM-\beta_{\mathrm{instr}}=\frac{0.9\cdot \lambda }{D_{\mathrm{Sch}}\cdot \cos(\theta )}$$
where $\beta_{\mathrm{physical}}$ is the value of the physical broadening of the X-ray reflection at its half-height according to the Scherrer formula; $\beta_{\mathrm{instr}}$ is the value of the instrumental broadening of the X-ray reflection; λ is the wavelength of the radiation; $D_{\mathrm{Sch}}$ is the size of the CSR calculated by the Scherrer method; θ is the angle of incidence of X-ray radiation.

However, this method of assessing CSR ($D_{\mathrm{Sch}}$) is suitable for samples in which no microstrains are observed. In the case of GFs that have undergone rolling machining, it is better to calculate the CSR ($D_{W-H}$) by a method that takes into account the microdistortions of the crystal lattice. It was developed by Williamson and Hall [54,55]:(4)$$\beta_{\mathrm{physical}}=FWHM-\beta_{\mathrm{instr}}=\frac{0.9\cdot \lambda }{D_{W-H}\cdot \cos\left(\theta \right)}+\beta_{\mathrm{stress}}$$
(5)$$\beta_{\mathrm{stress}}=\varepsilon \cdot 4\mathrm{tg}(\theta )$$
(6)$$\beta_{\mathrm{physical}}\cdot \cos\left(\theta \right)=\frac{0.9\cdot \lambda }{D_{W-H}}+\varepsilon \cdot 4\sin(\theta )$$
where $\beta_{\mathrm{physical}}$ is the value of the physical broadening of the X-ray reflection at its half-height according to the Williamson-Hall formula, $\beta_{\mathrm{instr}}$ is the value of the instrumental broadening of the X-ray reflection at its half-height, λ is the wavelength of radiation, D is the size of the coherent scattering regions, θ is the angle of incidence of X-ray radiation, $\varepsilon_{\mathrm{stress}}$ is the microstrains caused by machining processes, and $\beta_{\mathrm{stress}}$ is the value of broadening at the half-height of the reflex arising from microstrains.

To obtain the result, the dependence of $\beta_{\mathrm{physical}}\cdot \cos\left(\theta \right)$ on sin(θ) was plotted graphically, then approximated by a linear function and extrapolated to intersect with the ordinate axis. At the intersection point, the size of the CSR was calculated, and the values of microstrains were determined by the tangent of the slope angle.

### 2.7. Investigation of the Nanocrystallites Misorientation Angles in Graphite Foils

Also, to determine the structural perfection of the GFs, the angles of misorientation (*φ*) of nanocrystallites [56] were determined by X-ray: the rocking curves for the diffraction reflection of hexagonal graphite (00.6) were recorded on the Rigaku Ultima IV diffractometer when focusing Bragg-Brentano using CuKα = 0.15425 nm radiation. The survey was carried out at a fixed value of the angle 2*θ*, which corresponds to the reflection (00.6) when scanning at the angle *θ* from 10° to 80° in the diffractometer mode *θ*.

Subsequent determination of the angle of misorientation of nanocrystallites (*φ*), in which the reference planes (00.1) are parallel to the investigated surface of the foil samples, was determined as follows:(7)$$\varphi =\frac{FWHM}{2}$$

### 2.8. Preparation of Flexible Graphite Foils Samples for the Study of Mechanical Properties

Strips were cut out from each GFs 365, 366, 367 with densities of 0.7, 1.0, 1.3, 1.6, 1.75 g/cm^3^ in the direction of rolled products (along *L*) and across it (along *T*) with dimensions of 150 × 25 mm (Figure 4) for the tensile tests by the standard ASTM F152-95 [57].

To test the tensile strength of the material, 10 samples were cut in each direction (summary: 150 units along *L* and 150 units along *T*). In total, 15 types of tested materials were obtained, each of which was studied in 10 samples for the accumulation of statistics.

Studies of the ultimate strength of GFs 365, 366, 367 with densities of 0.7, 1.0, 1.3, 1.6, 1.75 g/cm^3^ were carried out on a universal testing machine HxK-S/U modification H5K-S with a break tooling (Figure 5) at a tensile rate of 10 mm/min. Deviations in the measurement of force and displacement were not more than 0.5%.

The elastic performance tests from GF’s sheets were carried out according to the standard ASTM F36-15 [58]. One of these characteristics is compressibility—this is a property of the material that reflects the work of elastic and plastic deformations under compression load, measured as the ratio of the thicknesses after compression and before it. Another elastic characteristic is recoverability—the property of the material that reflects the operation of the elastic component in relation to the total deformation. Resiliency is a property of a material that displays the operation of elastic forces after unloading. Figure 6 is a schematic representation of a study of these characteristics.

For studies on elastic characteristics, GFs 365, 366, 367 with densities of 0.7, 1.0, 1.3, 1.6, 1.75 g/cm^3^ with dimensions 30 × 30 mm were cut out 5 samples for each position according to Table 1 (summary 75 units). The standard implies that the thickness of the samples should be at least 1.6 mm, therefore, the prefabricated samples were created for testing them on a universal HxK-S/U machine of modification H5K-S with the appropriate tooling (Figure 7), the total thickness of which exceeded the specified threshold. A steel indenter with a diameter of 6.4 mm was used. According to the standard, a preload of 0.7 MPa (22.2 N) was first applied to the samples so that the surfaces of the foils in the sample fit tightly together. Then the test was carried out at the main load of 34.6 MPa (1112 N), and the dependence of the load and unload on the movement was built (Figure 7b).

The calculation of compressibility, recovery, and resiliency was carried out according to Equations (8)–(10), respectively:(8)$$Compressibility=\frac{h-h_{c}}{h}\cdot 100\%,$$
(9)$$Recovery=\frac{h_{r}-h_{c}}{h-h_{c}}\cdot 100\%,$$(10)$$Resiliency=\frac{h_{r}-h_{c}}{h_{c}}\cdot 100\%,$$
where, h is the initial thickness determined under the preload, $h_{c}$ is the critical thickness determined under the main load, and $h_{r}$ is the thickness determined after removal of the main load.

## 3. Results and Discussion

### 3.1. Structural Changes of the Graphite Matrix at Different Technological Stages

The production of GF is due to the various processing steps of purified natural graphite. The fractional composition of PG was studied by Cilas 1180 (Figure 8).

The fractional composition of purified natural graphite was equivalent to the distribution from Figure 1. This results consistency allows us to conclude that the graphite purification method presented in the article has little effect on changing the fractional composition of the raw material.

The PG diffractogram together with the standard sample is shown in Figure 9. Graphite was characterized by the presence of the main α-phase with the small content of β-phase. The quantitative analysis was not carried out in the work.

This α-graphite was characterized by the high intensity from basal planes of the (00.*l*) type, with significantly lower intensity values from prismatic and pyramidal ones. This indicates the high orientation degree of the basal planes along the flake plane, which is characteristic of natural graphite.

Accounting for the instrumental contribution to the full width at half maximum of the reflex ($\beta_{\mathrm{instr}}$) was carried out by recording the diffractogram (Figure 9) of the *LaB_6_* standard sample (Table 2) with similar survey parameters as for PG, GIC, OG, TEG, and GF. The determination of the full width at half maximum of this standard allows the assessment of the contribution of instrumental broadening, which makes it possible to find the physical broadening of the reflections of the studied samples. This physical broadening is solely attributed to the structural imperfection of graphite nanocrystallites caused by processing, while excluding the influence of the geometry of the X-ray diffractometer.

In each of them, graphite undergoes structural changes, for the study of which the X-ray phase and structural analysis method was used for each stage (Figure 9 and Figure 10).

Then the structural parameters of graphite nanocrystallites were calculated at different stages. The results are given in Table 3. It follows from the analysis of the table that natural graphite has strongly pronounced values of reflex intensity from the basic norms, as well as high CSR values. The difference in the values of the methods for determining the CSR is associated with high values of compressive microstrains in the PG along the baseline normal, presumably due to its pre-treatment.

The calculation of structural parameters for GIC and OG was difficult. The more thorough study of structural transformations for the graphite intercalated compound and oxidized graphite is needed, which would be better separated into the independent study.

Treatment of natural graphite with fuming nitric acid results in the formation of graphite nitrate, which intercalates between the graphite layers. The step number, that is, the number of graphene layers (00.*l*) between the two nearest layers of the intercalate, is determined by the XRD method (CIG, Figure 10, Table 3). The presence of the main stage III (highlighted in light green in Table 3 for CIG) and impurity stage II (highlighted in light orange in Table 3 for CIG) with identity periods of 1.454 nm and 1.119 nm, respectively, was revealed. Flushing with water led to the deintercalation of nitric acid and the formation of a non-stoichiometric adduct—OG (Figure 10) with the residual stage V (highlighted in blue in Table 3 for OG).

After the intercalation and washing of GIC, the weight gain (the ratio of the mass of OG to the mass of natural graphite used) was 7.7 ± 1.0%. Then, sharp heating of OG was carried out to a temperature of 900 °C in the air, which led to the formation of a dispersing pressure inside the particle, tearing it from the inside along the base layers, significantly changing the structure of graphite with the formation of TEG (Figure 10). TEG is quite difficult to study by X-ray methods, however, it is possible to calculate the parameters along the baseline (00.2). CSR’s calculations were carried out only using the Scherrer method—there was the significant decrease in the nanocrystallites size along the c-axis, associated with the OG’s thermal expansion. The bulk density of the TEG was 4.9 ± 0.2 g/L, the yield of the solid product (the ratio of the mass of the obtained TEG to the mass of the used exhaust gas) was 85.0 ± 0.5%, and the carbon yield (the ratio of the mass of the obtained TEG to the mass of the used PG) was 92 ± 1%.

The morphology of PG, OG, TEG flakes and GF was shown in Figure 11.

It is not possible to examine an intercalated graphite compound using an electron microscope because it is highly chemically reactive, which can cause equipment failure. The SEM images shown indicate that PG (Figure 11a) does not change its macrostructure during the stages of acid intercalation (production of GIC) and subsequent washing and drying of the OG’s raw material (Figure 11b). During heat treatment, the scales "expand" and form TEG (Figure 11c). The flat surface area of the particles remains virtually unchanged, but the thickness increases several hundred and sometimes thousands of times. Based on the nature of graphite, the process of obtaining TEG, and the values of CSR $D_{\mathrm{Sch}}$ (Table 3, for PG and TEG), it follows that nanocrystallites undergo destruction and grinding (decrease from 57.3 nm to 20.5 nm). Changes in the parameters of this process can also affect the final properties of the finished product. In our study, this parameter was kept constant.

Using cold rolling TEG was produced GFs (Figure 11d). The scratches were found on GFs by during rolling. Obviously, this negatively affects the strength characteristics of the material. Then the TEG was pressed to the GF with the density of 0.7 g/cm^3^, but of different thicknesses (see Table 1).

### 3.2. Structural Changes in Flexible Graphite Foils

According to the XRD (Figure 12), GFs 365, 366, 367 with densities of 0.7, 1.3, and 1.75 g/cm^3^ is graphite in its crystalline structure. Reflections from the reference plane and the corresponding parameters were calculated using the methods of profile analysis in the PDXL2 program and recorded in Table 4.

From the analysis of the data, it can be seen that the integral intensity of reflection from the reference planes increases as the density increases. This effect indicates that the porosity of the GFs decreases and the volume fraction of the material increases, which is involved in interaction with X-rays. In addition, the signal is amplified due to the fact that more base planes enter the reflecting position as a result of the rotation of nanocrystallites during rolling.

The dimensions of the CSR are preferably those calculated according to the Williamson-Hall method, since the GFs have been machined, which entails an increase in crystal lattice distortions inside the crystallites. The analysis of CSR revealed an increase in the size of nanocrystallites with an increase in the density of GFs. This is because at high loads that occur inside nanoscale grains, they coagulate because enough energy accumulates for the formation of Van der Waals bonds between the surface planes of coaxial crystallites. Also, the increase in the integral intensity of reflections from the basal planes may indicate coagulation of graphite nanocrystallites in the foils.

The increase in microstrains (Table 4) in the studied samples is associated with an increase in the applied load by the shafts during the rolling of graphite foils. From this, the following trend can be identified: when rolling foils, the sizes of nanocrystallites will increase, however, the distortion amount of their crystal lattice will also increase. The latter can reduce the physical and mechanical properties of foils. In our study, we apparently did not reach such densities of the graphite foils to notice this decrease.

However, we can assume the following pattern: at graphite foil densities from 0.7 g/cm^3^ to 1.75 g/cm^3^, the effects of declining foil porosity and coagulation of nanocrystallites are the prevailing factors in describing the properties of materials. Compared to them, the influence of microdistortions of the crystal lattice on the characteristics of the foil is insignificant in this case.

### 3.3. Tensile Strength of Flexible Graphite Foils

Regardless of the GFs thickness with increasing density, the tensile strength increased significantly, while all samples with a density of 0.70 g/cm^3^ were characterized by approximately the same values of the tensile strength, which were about 3.1 MPa. The graphs obtained on a tensile machine for GFs 365, 366, 367 of this density are shown in Figure 13.

Also, from the analysis of the graphs obtained during the tensile tests of foils of different densities (Figure 14), there is a change in the tensile strength depending on the density in and across the rolling direction. The dependence of the tensile strength on the density of GFs is linear. When the density reaches ~1.4 g/cm^3^, the slope of the strength versus density trend line changes, after which the dependence retains a linear character.

The phenomenon of increasing the tensile strength is explained as follows: tensile work is the force that must be overcome in order to move the TEG particles relative to each other, while this value is set by the area of contact between the particles and the coefficient of friction. As the density increases, the contact area increases, which explains this phenomenon. The change in the angular coefficient of dependence is associated with the collapse of pores in the material, which makes an additional contribution to the increase in the contact area and is confirmed by the results of porosimetry.

In comparisons of the obtained dependencies with each other, it should be noted that when switching from a row of foils with the largest to the average thickness, there is a slight increase in the tensile strength. However, when switching to a row with the smallest thicknesses, a significant drop in strength is clearly recorded, in the extreme case, a decrease to 12%. It is assumed that the reason for this was the ash impurities inevitably contained in natural graphite and remaining in the process of obtaining GF. Foreign impurities are stress concentrators that reduce the cross-sectional area of the material and can significantly reduce the tensile strength.

To study the composition of impurities in the GFs, their ash residue was obtained, which amounted to 0.24% of their weight. The fractional composition of the ash was studied similarly to natural graphite by the Cilas 1180 LD laser analyzer (Figure 15).

Less than 50% by weight of the particles had a size < 30 µm, but 3% with a size of 90–112 µm were present. The size of these large particles is comparable to the thickness of a series of thin foils with a high density of 1.75 g/cm^3^, that is, the particles are able to occupy the entire thickness of the foil, creating the highest stress concentrations relative to foils with a greater thickness.

### 3.4. Elastic Characteristics of Flexible Graphite Foils

With an increase in density from 0.7 g/cm^3^ to 1.75 g/cm^3^, there is a decrease in compressibility, an increase in recoverability, and, accordingly, a slight decrease in resiliency (Figure 16). At the same time, the thickness of the GFs practically does not affect the elastic characteristics.

The compressibility is responsible for the ability of a material to reduce its thickness under load. The graph (Figure 16a) shows that the dependence of the characteristic on the density is linear for all GFs. The recovery, as can be seen from Equation (10), is the ratio of the “rebound” value to the compression value. Its dependence on the density of GFs (Figure 16b) is described by a polynomial function of the second degree and is the same for all samples. Resiliency is the ability of a material to regain part of its thickness after compression. The decrease in elasticity is logical, since it is associated with a decrease in the ability of the material to be compressed with increasing density. At the same time, the nature of its dependence on the density of the foil is described by polynomials of the second degree and differs between GFs with different thicknesses (Figure 16c).

This is due to the fact that compressibility and recoverability are more dependent on the density (porosity) of GFs, while the elastic characteristics are also significantly affected by the structural state of nanocrystallites, which change during the rolling of foils.

### 3.5. Anisotropy of Structural Characteristics and Physical and Mechanical Properties of Flexible Graphite Foils

#### 3.5.1. Forced Reorientation of Nanocrystallites

During rolling, structural changes occur in the graphite matrix. Two factors can be distinguished that affect the properties of the foil: these are changes in its density, and, as a dependent parameter, porosity, which characterizes how tightly the graphite particles adhere to each other, as well as the misorientation of graphite nanocrystallites in the direct material. These two processes will determine the physical and mechanical properties of GFs.

To study the structure of nanocrystallites, an analysis of TEM images was carried out (Figure 17). It follows that there is a significant porosity in the sample, which could not be estimated by this method, and the angles between the graphite flakes were calculated. It can be seen that the angles in denser foils are much smaller.

It was found that to further identify the deformation mechanism of flexible graphite foils, local mechanical properties should be studied [59]. Their values can vary significantly at different places on the surface [60].

#### 3.5.2. Misorientation Angles of Nanocrystallites

Since the analysis of the misorientation of nanocrystallite nodes by the TEM method is local, the X-ray technique of recording rocking curves by (00.6) reflection (Figure 18) was used to increase the statistical significance of the results. The full width at half-maximum (*FWHM*) of the resulting curves was calculated for GFs 365, 366, 367 with densities of 0.7, 1.3, and 1.75 g/cm^3^.

The analysis was performed on GFs 365, GFs 366, and GFs 367 of all thicknesses, with densities of 0.7, 1.3, and 1.75 g/cm^3^. The results are shown in Figure 19.

It follows from the analysis of the graphs that with an increase in the density of GF’s samples, the angles of misorientation of nanocrystallites decrease due to two processes: a decrease in porosity and coagulation of crystallites due to mechanical processing. A decrease in the angles of misorientation is also observed most strongly in thin foil (GFs 365). This confirms the assumption about the duality of the mechanism of deformation of the GF, since, other things being equal (the same density, similar dimensions of the CSR, and hence the porosity), this process can only be due to a decrease in misorientation.

#### 3.5.3. Tensile Strength of Flexible Graphite Foils

For one density and different directions of tension—along the rolling axis and in the direction at an angle of 90° relative to the axis, an increase in anisotropy is observed with increasing density (Figure 20).

When rolling GF to the maximum represented values of densities (1.75 g/cm^3^), the material is elongated by 2.7%, and the broadening is only 1.1%. The forces acting on the material have a dual character, first, the material is able to significantly decrease in thickness—to be pressed, second, the greater the amount of compression, the greater the contribution of extrusion to the process. At the same time, extrusion is about 2–3 times higher along the rolling direction. It can be assumed that with such a “pulling” of the material, the orientation of the TEG particles, that is, the orientation of the contact areas, also occurs. In this case, the material oriented along the rolling direction will exhibit a greater tensile strength than across the rolling, which is observed. The rows obtained from the original foil with a thickness of 0.76, 0.5, and 0.28 mm at a density of 1.75 g/cm^3^ show a similar anisotropy of 10–12%. Although at an initial density of 0.70 g/cm^3^, anisotropy, taking into account the statistical spread, is not observed, which indicates a small contribution of this factor at relatively low densities.

A more intuitive dependence of tensile strength on thickness is shown in Figure 21. It can be seen that for low densities, the thickness does not significantly affect the strength, while for high densities, there is a slight increase in the desired value during the transition from the largest thicknesses to the average ones and a significant drop from the average thicknesses to the smallest ones. At the same time, a similar character is inherent in the material along and across the rolling direction.

In addition to the anisotropy of mechanical properties in the GF, there is a similar nature of the distribution of such a physical quantity as Young’s modulus (Figure 22). The data correlate well with each other.

Anisotropy of the ultimate tensile values relative to the measurement directions is observed (Figure 23), which manifests for the density of GF 0.7 g/cm^3^ and increases with increasing density. At a maximum density of 1.75 g/cm^3^, there is a slight decrease in the desired parameter relative to the density of 1.6 g/cm^3^.

The exception was a series of GFs with the smallest thicknesses, for which the maximum stretch along the rolling decreased with increasing density. Comparing the behavior of the material along and across the rolling direction, there was a significant anisotropy (relative to the average values at a density of 1.75 g/cm^3^) in relative elongation, which was 60% for a number of foils with the greatest thicknesses, when moving to the average, the anisotropy thickness was 260%, and the smallest thicknesses showed 230%.

Since the tested samples have the same length, the absolute values of elongation can be compared directly. Stretching of samples with the greatest thicknesses is 2–3 times higher. This is probably due to the large absolute volume of voids within the material available for particle shear. At the same time, across the rolling direction, the behavior of all tested samples obeys a similar dependence and significantly exceeds the first direction.

## 4. Conclusions

GFs were obtained from oxidized natural graphite with thicknesses from 0.11 mm to 0.75 mm and densities from 0.70 to 1.75 g/cm^3^. The structural transformations of nanocrystallites at each stage of foil production were studied. It was shown that in natural graphite, the regions of coherent scattering of nanocrystallites had a size of 57.3 nm, and in TEG, the dimensions of the regions of coherent scattering were reduced to 20.5 nm.

It was found that the compressibility with increasing density of GFs decreased from 65% to 9%, the recoverability increased from 5% to 60%, and the resiliency decreased from 10% to 6%. It was shown that the elastic properties of GFs were determined by the structural state of nanocrystallites.

An increase in the anisotropy of the physical and mechanical properties of GFs was found with an increase in density along *L* and *T* up to 12% during mechanical tensile tests. These data are congruent with the anisotropy of the misorientation angles of the GF’s nanocrystallites, which reaches 11%.

The dimensions of the coherent scattering regions in nanocrystallites increase from 24.8 nm (ρ = 0.7 g/cm^3^) to 49.6 nm (ρ = 1.75 g/cm^3^), which indicates the coagulation of coaxial crystallites due to an increase in deformations caused by rolling.

It has been revealed that microstrains of the crystal lattice of GFs grow with increasing density. It is assumed that at densities from 0.7 to 1.75 g/cm^3^ the influence of such lattice distortions is insignificant, because at such density values, the predominant deformation mechanism is the decrease in the porosity of the material with simultaneous coagulation of nanocrystallites.

For further research, it is planned to study the thermal conductivity and electrical conductivity of these graphite foils for their use as a material for radiators, thermal interfaces and thermal conductors.

## Figures and Tables

**Figure 1 nanomaterials-14-00540-f001:**
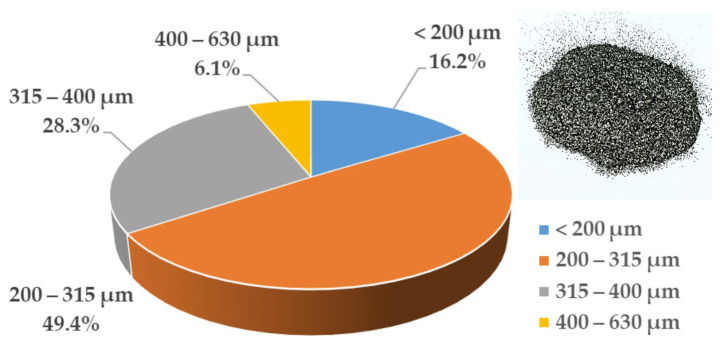
The fractional composition of the natural graphite.

**Figure 2 nanomaterials-14-00540-f002:**
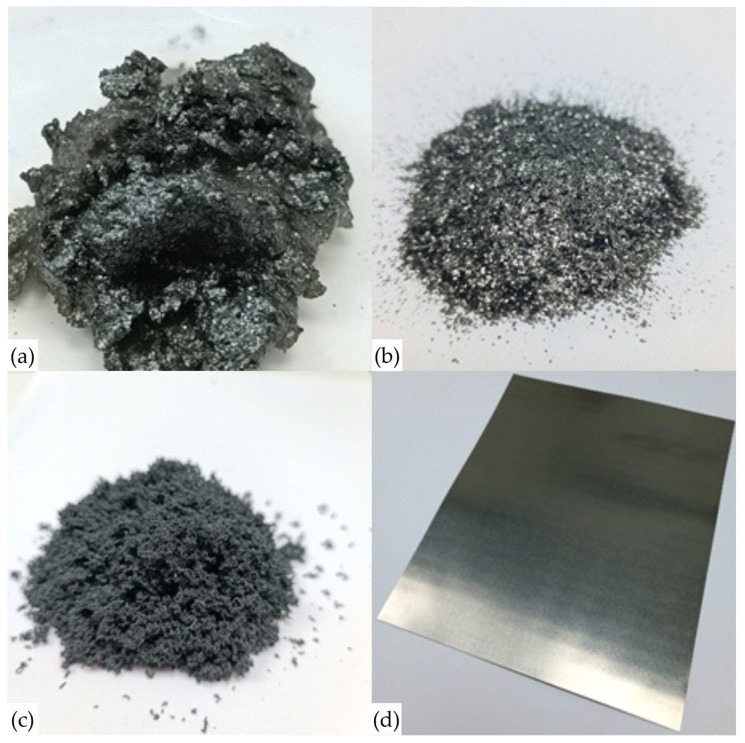
The image of graphite intercalation compound (**a**), oxidized graphite (**b**), thermally expanded graphite (**c**) and flexible graphite foil (**d**).

**Figure 3 nanomaterials-14-00540-f003:**
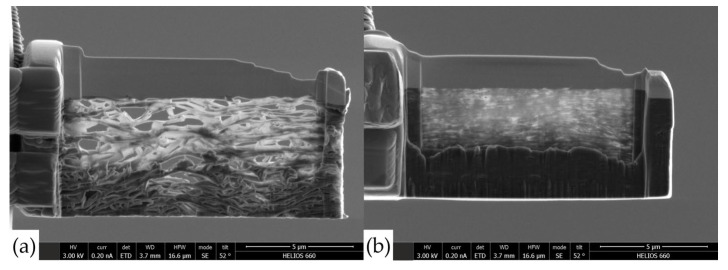
The preparation of the lamellas for the transmission electron microscope for GFs 367 with the density of 0.7 g/cm^3^ (**a**) and 1.75 g/cm^3^ (**b**).

**Figure 4 nanomaterials-14-00540-f004:**
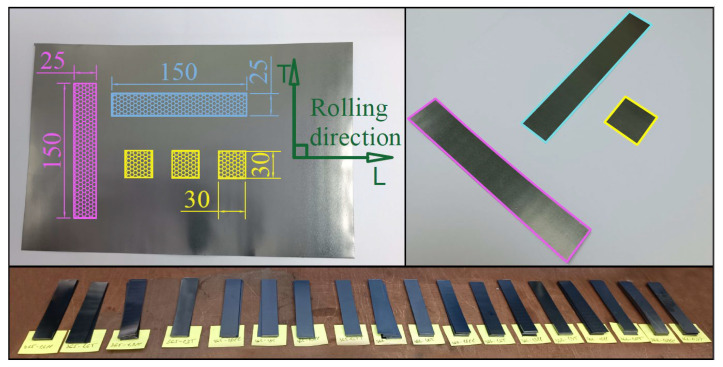
Scheme of cutting out samples for the tensile testing.

**Figure 5 nanomaterials-14-00540-f005:**
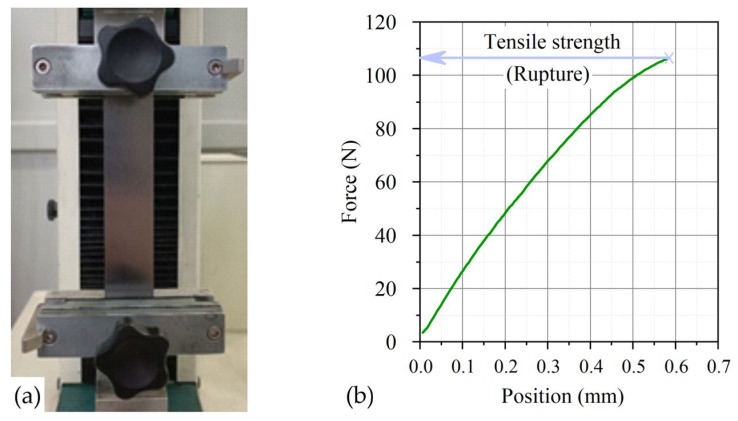
Image of the specimen (**a**) in the HxK-S/U test machine of H5K-S modification with tensile tooling and the graph of the dependence of the load on the movement during mechanical tensile tests of the foil (**b**).

**Figure 6 nanomaterials-14-00540-f006:**
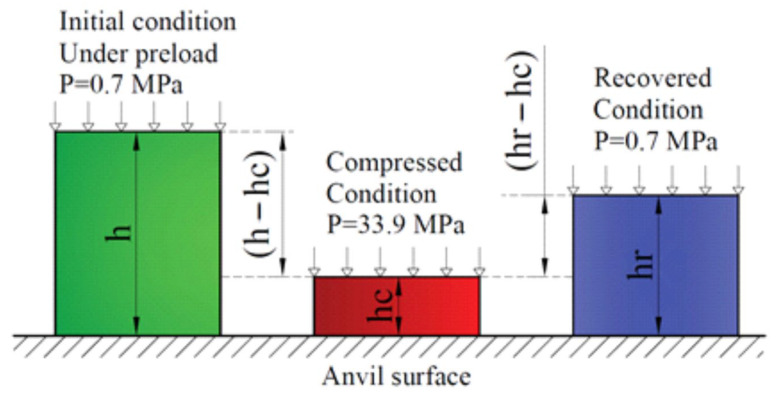
Scheme for investigation of the compressibility, recovery, and resiliency.

**Figure 7 nanomaterials-14-00540-f007:**
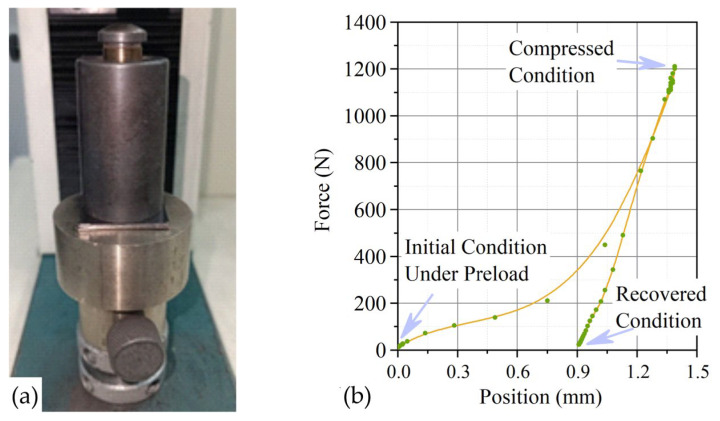
Image of the specimen (**a**) in the HxK-S/U test machine of modification H5K-S with equipment for testing elastic characteristics and a graph of the dependence of the load on the movement during mechanical tests of the foil (**b**) in determining the elastic characteristics.

**Figure 8 nanomaterials-14-00540-f008:**
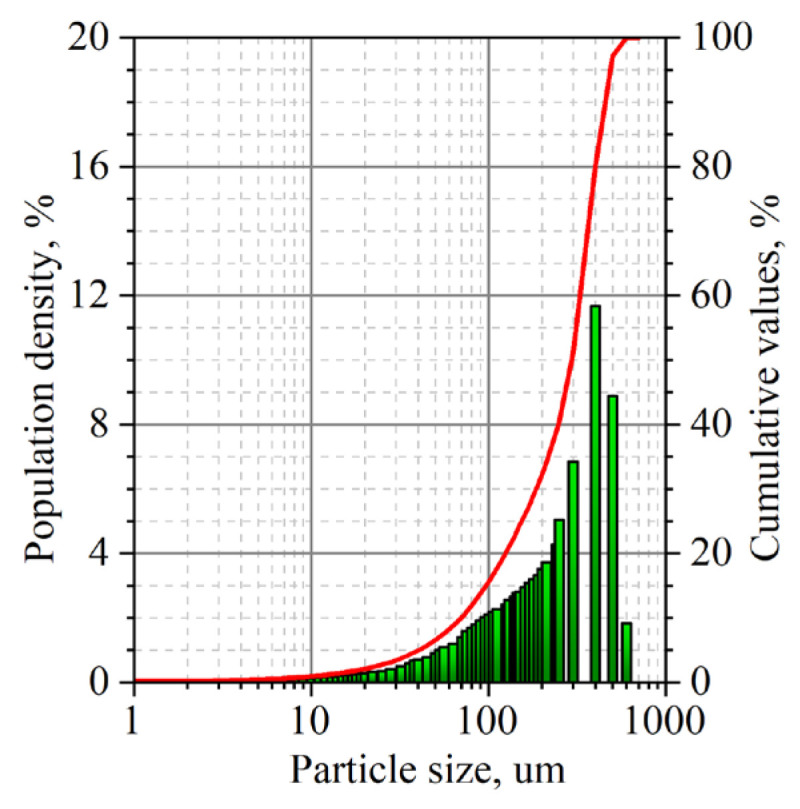
The fractional composition of natural graphite after additional purification.

**Figure 9 nanomaterials-14-00540-f009:**
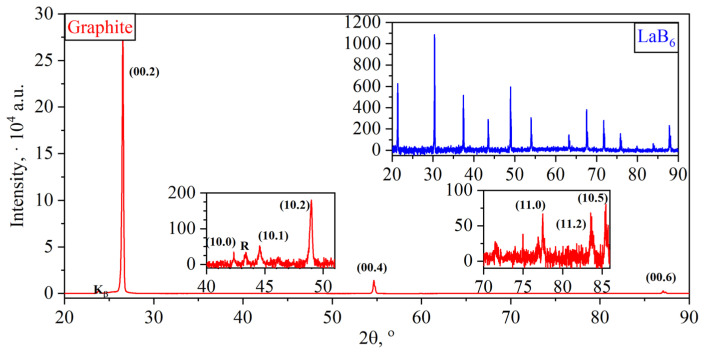
The diffraction pattern of natural graphite and the standard sample (*LaB*_6_).

**Figure 10 nanomaterials-14-00540-f010:**
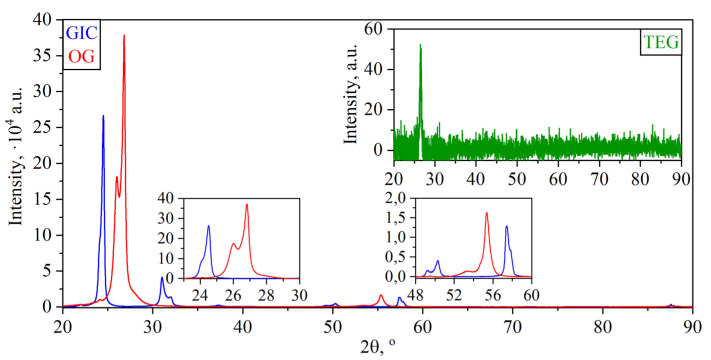
Diffractograms of GIC (**blue**), OG (**red**) and TEG (**green**).

**Figure 11 nanomaterials-14-00540-f011:**
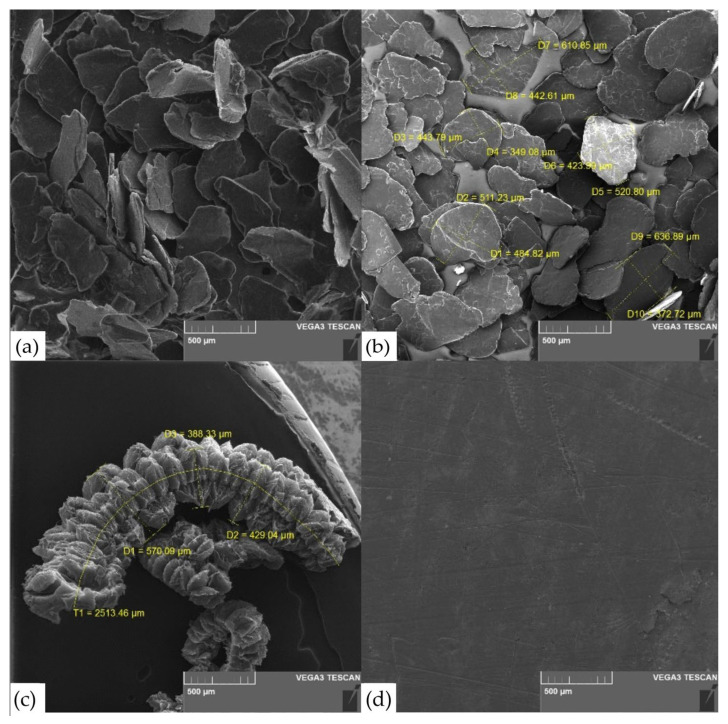
SEM image of PG (**a**), OG (**b**), TEG (**c**), GF (**d**).

**Figure 12 nanomaterials-14-00540-f012:**
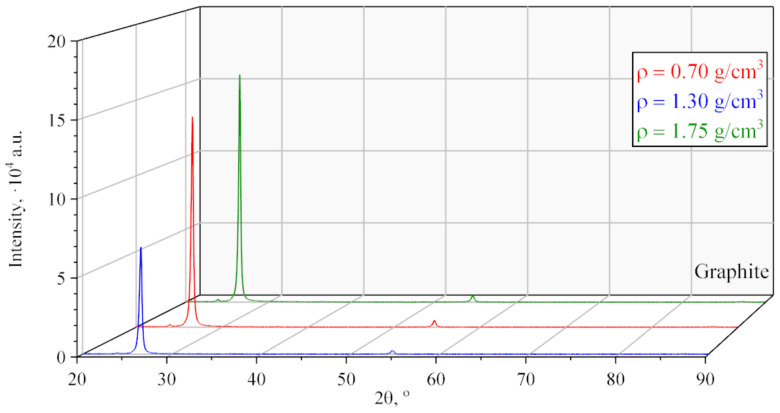
Diffractograms of GFs of different densities.

**Figure 13 nanomaterials-14-00540-f013:**
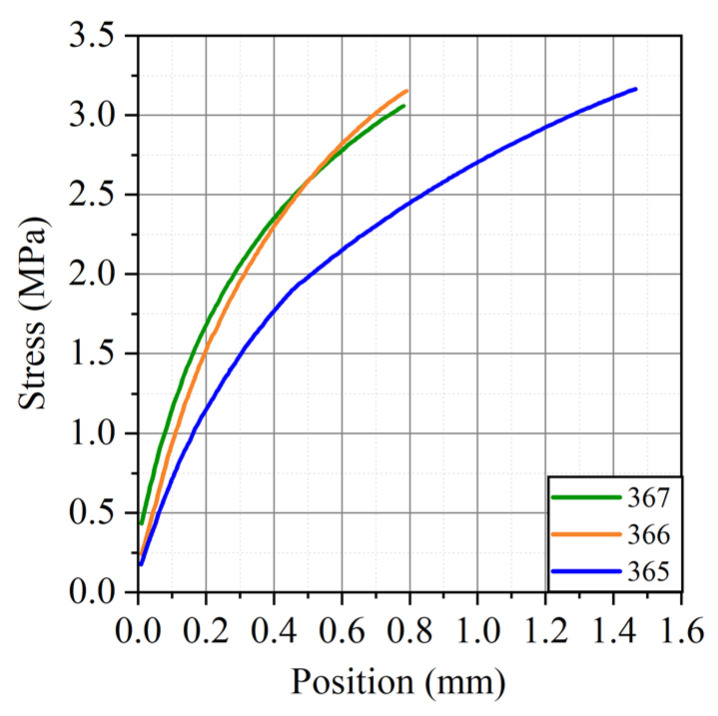
Graph of load versus displacement during tensile tests for GF 365, GF 366, and GF 367 with a density of 0.7 g/cm^3^ along the rolling direction.

**Figure 14 nanomaterials-14-00540-f014:**
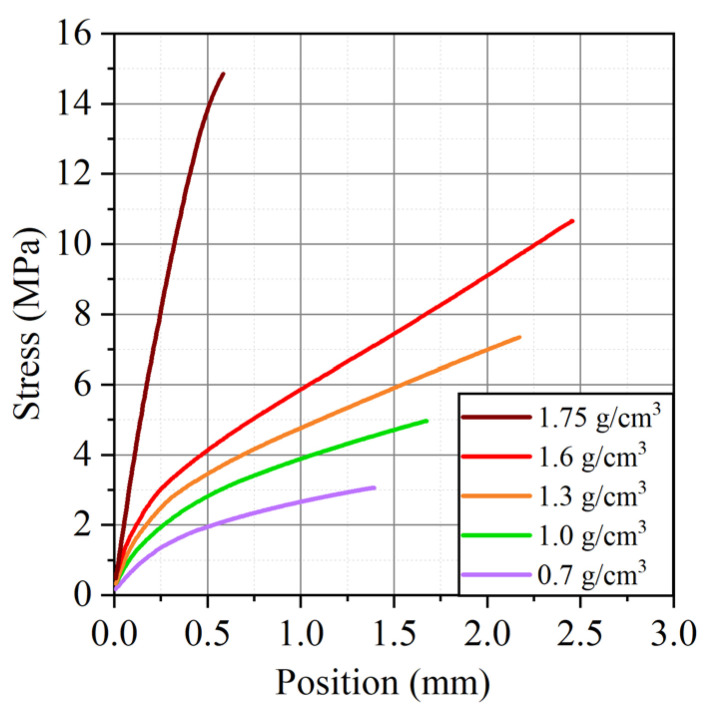
Dependence of the load on the displacement for different densities of GFs 367 along the rolling direction.

**Figure 15 nanomaterials-14-00540-f015:**
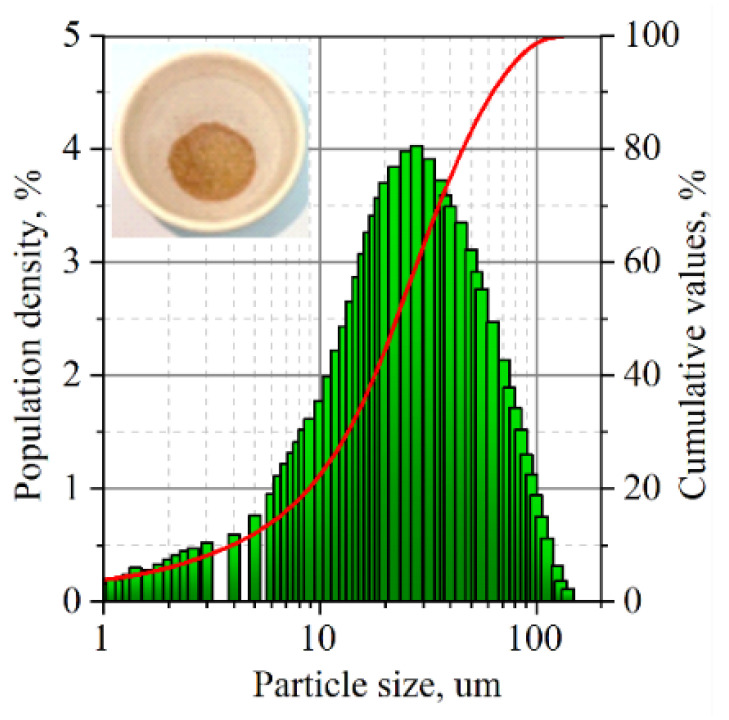
Particle size distribution of ash.

**Figure 16 nanomaterials-14-00540-f016:**
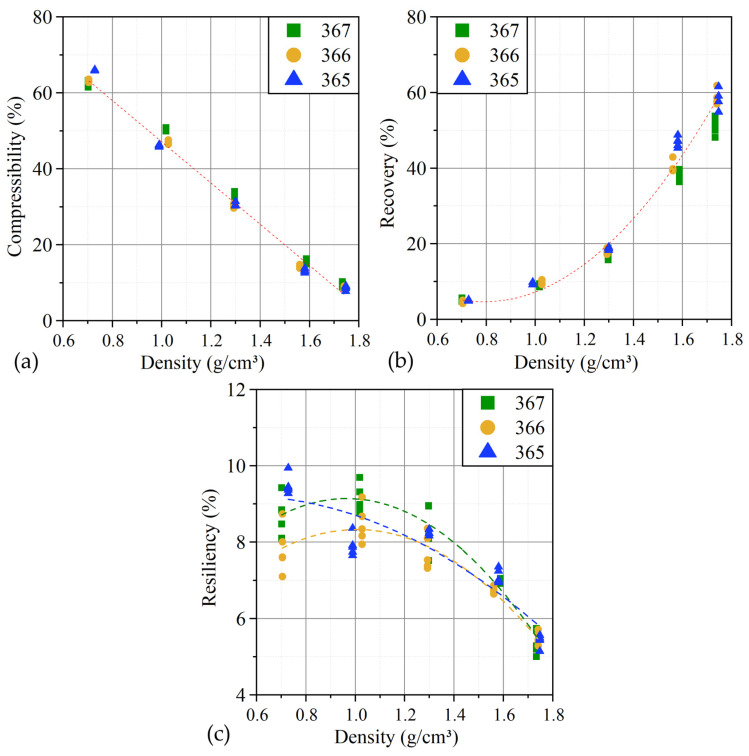
The relations of the GFs elastic characteristics (compressibility (**a**), recoverability (**b**), and resiliency (**c**)) on thickness and density.

**Figure 17 nanomaterials-14-00540-f017:**
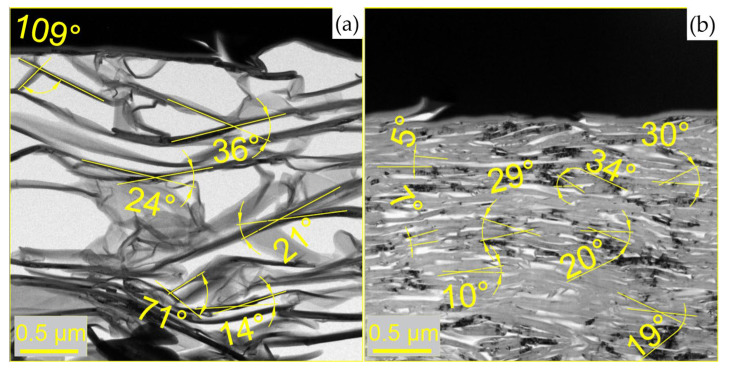
TEM images of GFs 367 with densities of 0.7 g/cm^3^ (**a**) and 1.75 g/cm^3^ (**b**).

**Figure 18 nanomaterials-14-00540-f018:**
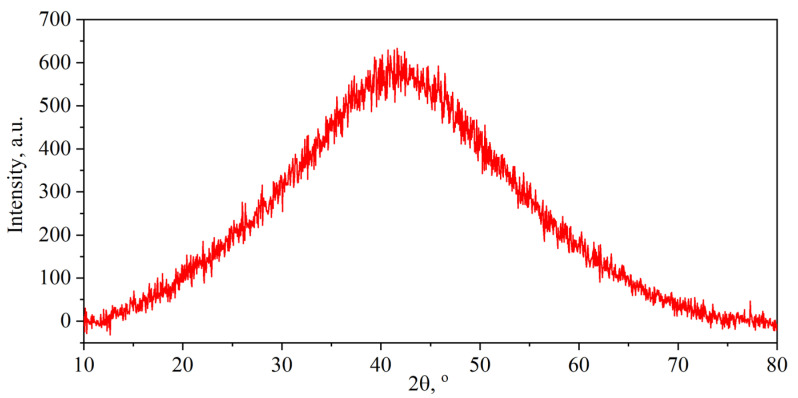
Rocking curve of GF 365 with a density of 0.7 g/cm^3^.

**Figure 19 nanomaterials-14-00540-f019:**
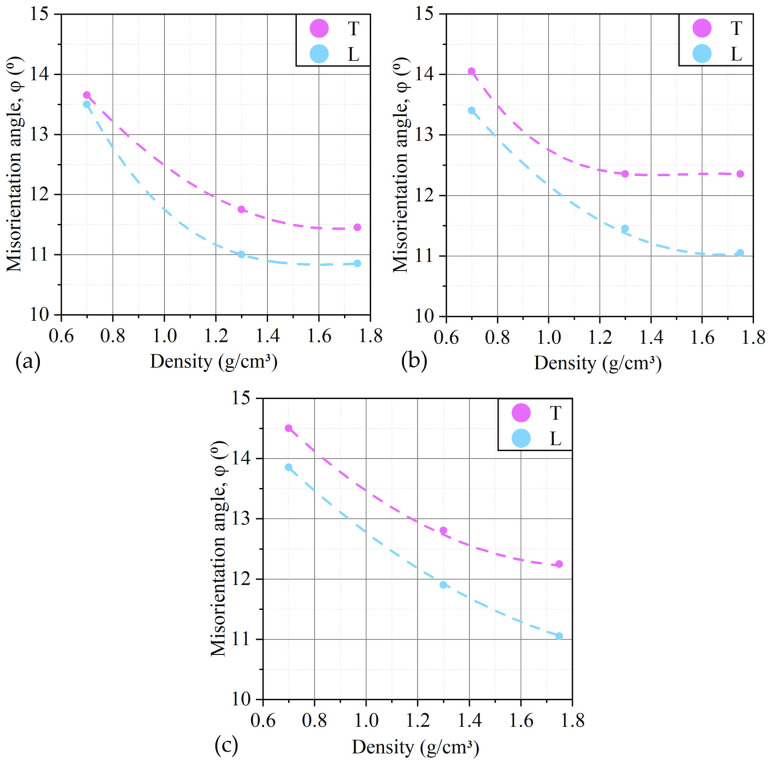
Distribution of the values of the misorientation angles of nanocrystallites with the reflection (00.6) of GFs 365 (**a**), GFs 366 (**b**), and GFs 367 (**c**) along the rolling direction (*L*) and across it (*T*).

**Figure 20 nanomaterials-14-00540-f020:**
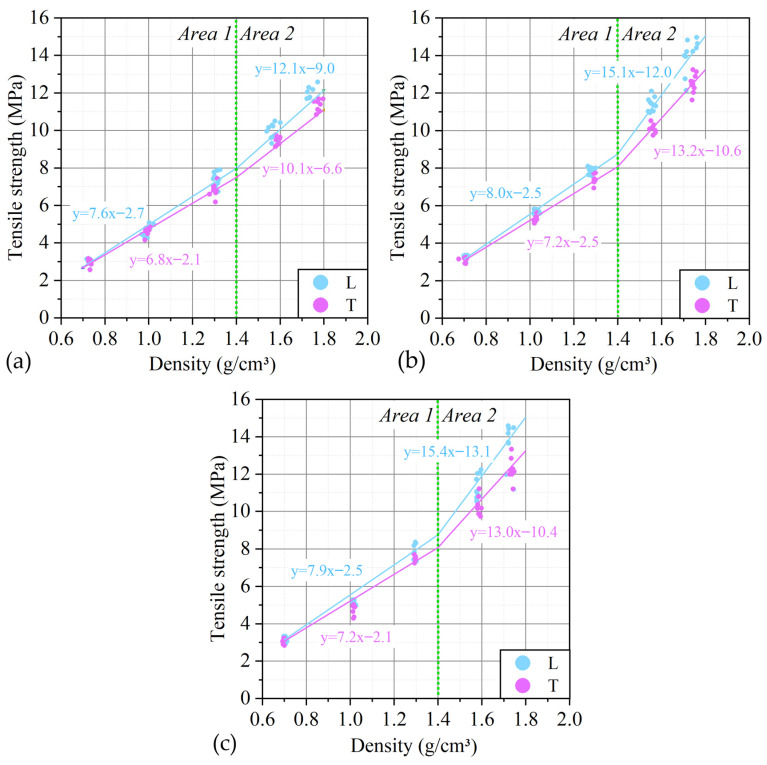
The dependence of the tensile strength on the density of GF 365 (**a**), GF 366 (**b**) and GF 367 (**c**) and the direction of rupture for the tested samples.

**Figure 21 nanomaterials-14-00540-f021:**
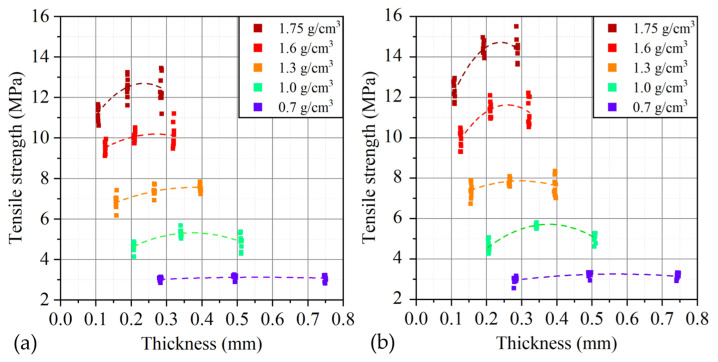
Dependence of the ultimate strength of the GF on its thickness and density along (**a**) and across (**b**) the rolling direction.

**Figure 22 nanomaterials-14-00540-f022:**
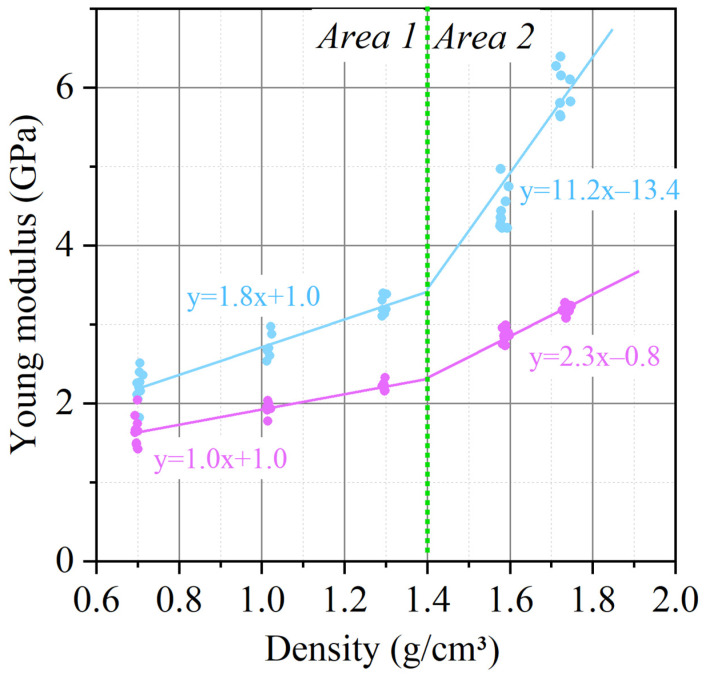
Anisotropy of Young’s modulus depending on the density of GF 367.

**Figure 23 nanomaterials-14-00540-f023:**
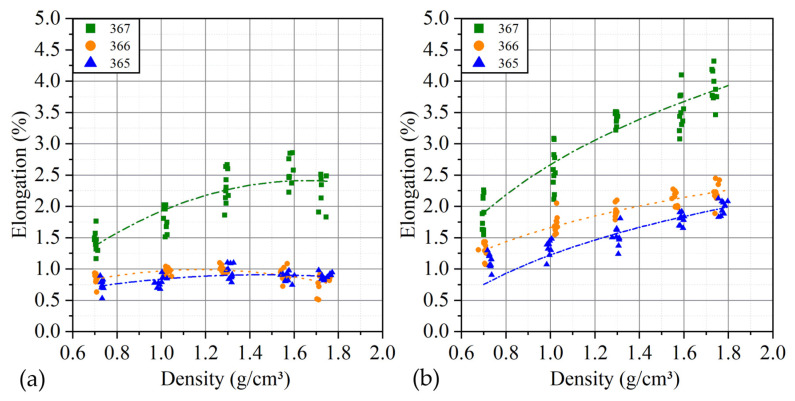
Dependence of GF tensile strength on density along *L* (**a**) and across it (along *T*) (**b**).

**Table 1 nanomaterials-14-00540-t001:** Parameters of the tested GFs samples.

GF 365		GF 366		GF 367	
h, μm	ρ, g/cm^3^	h, μm	ρ, g/cm^3^	h, μm	ρ, g/cm^3^
282 ± 5	0.73 ±0.01	494 ± 7	0.705 ± 0.03	746 ± 8	0.70 ± 0.01
207 ± 3	0.99 ± 0.03	342 ± 3	1.03 ± 0.07	510 ± 5	1.02 ± 0.01
157 ± 5	1.30 ± 0.03	266 ± 4	1.29 ± 0.11	396 ± 5	1.30 ± 0.01
127 ± 5	1.58 ± 0.04	211 ± 4	1.56 ± 0.02	320 ± 4	1.59 ± 0.01
108 ± 3	1.75 ± 0.04	191 ± 6	1.74 ± 0.03	286 ± 8	1.73 ± 0.02

**Table 2 nanomaterials-14-00540-t002:** Structural characteristics of *LaB_6_*.

2*θ*, °	*d*, nm	$\beta_{\mathrm{instr}}$, °	hk.l
21.33	0.41623 (7)	0.079 (6)	100
30.36	0.29416 (3)	0.078 (3)	110
37.43	0.24005 (2)	0.085 (4)	111
43.49	0.20790 (3)	0.078 (6)	200
48.95	0.18592 (1)	0.085 (3)	210
53.99	0.16969 (2)	0.081 (5)	211
63.22	0.14697 (1)	0.071 (8)	220
67.55	0.13856 (1)	0.087 (3)	300
71.75	0.13145 (1)	0.078 (4)	310
75.86	0.12532 (1)	0.091 (7)	311
79.91	0.11995 (2)	0.100 (20)	222
83.87	0.11527 (1)	0.093 (17)	320
87.81	0.11108 (1)	0.093 (7)	321

**Table 3 nanomaterials-14-00540-t003:** Structural parameters of PG, GIC, OG, and TEG.

Graphite Type	2*θ*, °	*d*, nm	*FWHM*-$\beta_{\mathrm{instr}}$, °	*I*_int_, Count·°	$D_{\mathrm{Sch}}$, nm	$D_{W-H}$, nm	*ε*	hk.l
PG	26.54	0.33557 (4)	0.191 (2)	51,320	57.3 ± 13.3	36.5 ± 0.9	−0.0027	00.2
	48.96	0.18590 (5)	0.203 (15)	32				10.2
	54.66	0.16776 (7)	0.157 (3)	2992				00.4
	59.81	0.15451 (19)	0.790 (12)	34				10.3
	87.07	0.11181 (1)	0.159 (2)	723				00.6
GIC	24.07	0.36948 (10)	0.644 (5)	45,393	-	-	-	00.3
	24.50	0.36304 (3)	0.261 (3)	58,576				00.5
	30.99	0.28829 (5)	0.421 (5)	20,275				00.6
	32.03	0.27917 (8)	0.730 (20)	9624				00.4
	37.28	0.24097 (7)	0.790 (20)	2451				00.6
	40.43	0.22294 (17)	0.780 (50)	495				00.5
	49.15	0.18523 (4)	0.400 (30)	834				00.6
	50.28	0.18133 (4)	0.492 (14)	3034				00.8
	57.38	0.16054 (1)	0.264 (4)	4492				00.9
	57.87	0.15920 (1)	0.480 (20)	2858				00.7
	66.09	0.14125 (6)	0.970 (17)	81				00.8
	77.25	0.12340 (4)	0.780 (50)	228				00.9
	79.38	0.12061 (1)	0.860 (50)	241				00.12
OG	25.87	0.34409 (6)	1.334 (9)	150,868	-	-	-	
	26.81	0.33229 (6)	0.299 (6)	81,535				00.2
	53.25	0.17188 (11)	1.430 (60)	1901				
	55.26	0.16610 (1)	0.606 (6)	9837				00.4
	87.37	0.11153 (1)	0.508 (14)	589				00.6
TEG	26.57	0.33520 (40)	0.410 (30)	13	20.5 ± 1.5	-	-	00.2

**Table 4 nanomaterials-14-00540-t004:** Structural characteristics of GFs.

GF	Thickness, µm	2*θ*, °	*d*, nm	*FWHM* -$\beta_{\mathrm{instr}}$, °	*I*_int_, Count °	$D_{\mathrm{Sch}}$, nm	$D_{Y-X}$	*ε*	hk.l
365	0.7	26.55	0.33547 (8)	0.336 (8)	20,282	24.8 ± 0.5	24.8 ± 1.3	0.0003	00.2
		54.65	0.16779 (2)	0.363 (7)	908				00.4
		86.94	0.11195 (1)	0.463 (19)	101				00.6
	1.3	26.56	0.33531 (8)	0.302 (8)	37,966	24.5 ± 4.7	37.5 ± 1.7	0.005	00.2
		54.67	0.16774 (2)	0.342 (7)	1662				00.4
		87.05	0.11185 (2)	0.587 (17)	184				00.6
	1.7	26.53	0.33567 (7)	0.285 (7)	40,690	24.0 ± 5.2	39.7 ± 1.5	0.0057	00.2
		54.65	0.16779 (2)	0.386 (8)	1805				00.4
		87.13	0.11177 (1)	0.590 (13)	201				00.6
366	0.7	26.54	0.33553 (7)	0.367 (7)	23,360	21.4 ± 3.0	27.2 ± 1.1	0.0035	00.2
		54.63	0.16787 (2)	0.390 (7)	1105				00.4
		87.06	0.11184 (3)	0.620 (20)	124				00.6
	1.3	26.55	0.33550 (7)	0.333 (8)	43,738	24.4 ± 1.5	26.7 ± 0.8	0.0015	00.2
		54.65	0.16781 (2)	0.360 (7)	2109				00.4
		87.07	0.11183 (1)	0.500 (2)	274				00.6
	1.7	26.55	0.33551 (3)	0.288 (2)	60,881	23.3 ± 6.1	49.6 ± 1.9	0.0078	00.2
		54.67	0.16773 (2)	0.379 (9)	2812				00.4
		87.09	0.11181 (2)	0.670 (20)	356				00.6
367	0.7	26.56	0.33530 (6)	0.381 (7)	25,849	20.7 ± 2.1	24.8 ± 1.0	0.0027	00.2
		54.67	0.16775 (2)	0.418 (7)	1265				00.4
		87.04	0.11186 (3)	0.610 (20)	167				00.6
	1.3	26.55	0.33541 (7)	0.350 (7)	48,555	23.2 ± 1.3	25.2 ± 0.9	0.0012	00.2
		54.65	0.16781 (2)	0.379 (6)	2511				00.4
		87.08	0.11181 (1)	0.515 (13)	337				00.6
	1.7	26.53	0.33565 (8)	0.321 (8)	58,030	22.4 ± 4.1	33.9 ± 1.4	0.0052	00.2
		54.65	0.16781 (2)	0.392 (7)	2993				00.4
		87.068	0.11183 (2)	0.621 (17)	396				00.6

## Data Availability

The data presented in this study are available on request from the corresponding author.

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
