# Peer review of "The Influence of Machining Conditions on the Orientation of Nanocrystallites and Anisotropy of Physical and Mechanical Properties of Flexible Graphite Foils"

_nanomaterials, 2024, doi:10.3390/nano14060540_

Round 1

Reviewer 1 Report

Comments and Suggestions for Authors

The manuscript titled “The Influence of Machining Conditions on the Orientation of Nanocrystallites and Anisotropy of Physical and Mechanical Properties of Flexible Graphite Foils” by Shulyak, V.A.; et al. is a scientific work where the authors studied the performance of graphite foils at different nanocrystallite formation stages and the anisotropy of their mechanical properties was also monitored. This knowledge could serve to design more durable composite materials. The manuscript is generally well-written and this is a topic of growing interest.

However, it exists some points that need to be addressed (please, see them below detailed point-by-point) to improve the scientifc quality of the submitted manuscript paper before this article will be consider for its publication in Nanomaterials.

1) INTRODUCTION. This section clearly summarizes the state-of-the-art of the examined field. No actions are requested from the authors.

2) MATERIALS AND METHODS. “In the fourth stage, there was primary rolling (…) Detailed parameters are presented in Table 1” (page 3). What was the population size (N) taken into account to ascertain the parameters of the graphite foil samples?

3) “2.4. Examination of foil samples using SEM” (page 5). What was the acceleration electron voltage used to acquire the SEM images? Then, did the authors employ any contrast agent? In case affirmative, do the authors consider that it could be negatively affect to the data interpretation? Some information should be provided in these regards.

4) Finally the authors should consider to move the data linked to the Fig. 7 (page 6) and Fig. 8 (page 8) in the Results and Discussion section.

5) RESULTS AND DISCUSSION. Table 3 (page 11). Why did the authors no calculate all the structural parameters for all the examined samples? Same comment for the Table 4 (page 12). Some discussion should be provided in this regard.

6) Figure 17, X-axis (page 15). Please, the authors should modify the commas by points for the density values displayed in the X-axis. Same comment for the Figures 19-23 (pages 16-23).

7) “3.5. Anisotropy of structural characteristics and physical and mechanical properties of flexible graphite foils” Subsection “3.5.1. Forced reorientation nanocrystallites” (pages 15-16). Here, even if I agree with the data gathered by the authors and in order to strengthen the significance of this work, it should not be neglected the local mechanical properties [1] which can vary from graphite foil specific microstructure locations [2].

[1] Magazzù, A.; Marcuello, C. Investigation of Soft Matter Nanomechanics by Atomic Force Microscopy and Optical Tweezers: A Comprehensive Review. Nanomaterials 2023, 13, 963. https://doi.org/10.3390/nano13060963.

[2] Chen, Y.; Tu, C.; Liu, Y.; Liu, P.; Gong, P.; Wu, G.; Huang, X.; Chen, J.; Liu, T.; Jiang, J. Microstructure and mechanical properties of carbon graphite composites reinforced by carbon nanofibers. Carbon Lett. 2023, 33, 561-577. https://doi.org/10.1007/s42823-022-00445-4.

8) “3.5.2. Misorientation angles of nanocrystallites”. “The analysis was performed on GF 365, GF 366, and GF 367 of all thicknesses, with densities of 0.7, 1.3, and 1.75 g/cm3” (page 16). The significant figures should be homegenize. Please, the author should take this comment into account for the rest of the main manuscript body text.

9) CONCLUSION. This section perfectly remarks the most relevant outcomes found by the authors in this field. The authors should add a brief statement to discuss about the future line actions to pursue this research and the open perspectives.

Comments on the Quality of English Language

The manuscript is generally well-written albeit it may be desirable if the authors could do a final check in order to polish final details susceptible to be improved.

Reviewer 2 Report

Comments and Suggestions for Authors

This paper deals with The Influence of Machining Conditions on the Orientation of Nanocrystallites and Anisotropy of Physical and Mechanical Properties of Flexible Graphite Foils. This paper contains sound experimental results. However, some major revisions have to be done before the paper can be published in this journal.

1.      The formatting of the article needs to be strictly observed, the format within the text is somewhat chaotic, the language expression needs further improvement, meanings must be provided for characters or symbols when they appear or before their appearance, their usage also needs to comply with standards, and the size of tables needs adjustment.

2.      The introduction lacks a brief description of the methods and content of this study, is missing the purpose and significance of the research, and strict supplemental explanations are needed for this part.

3.      The content presentation should be appropriate, and the appearance of images and tables should be used to demonstrate or express a particular issue, rather than simply being listed haphazardly. It is recommended to reduce and consolidate the images in the paper.

4.      The methods are mixed up with the results and discussions. The methods should be used to describe the experiments conducted and how they were carried out, mainly focusing on the procedure. The results and discussions should analyze and interpret the experimental outcomes. In this article, the methods section contains an excessive list of results and data. It is recommended to consolidate and transfer all of them to the results and discussion section.

5.      What do the numbers 8, 9, 10 below Figure 10 represent?

6.      Please provide a detailed explanation of how to obtain the samples through Table 2, as stated in the first sentence below Figure 11.

7.      Are the commas used correctly in formulas 8, 9, and 10?

8.      It is preferable to describe the conclusions in a bullet-point format for a more structured presentation.

9.      Pay strict attention to the referencing format. Compare your format with articles in this journal to ensure compliance with the standards.

10.  The title of the article is too long; it should be properly refined and condensed.

Comments on the Quality of English Language

During the revision process, the language quality needs to be further improved by enhancing its logic and hierarchical structure.

Reviewer 3 Report

Comments and Suggestions for Authors

This work reports on the preparation of Graphite foils. Microstructures of the prepared graphite products were characterized, and the physical and mechanical properties were investigated.

However, this work is not suitable for publication in Nanomaterials. The description of "Materials and test Methods "is too long. In addition, this manuscript provides a brief description of XRD patterns and SEM micrographs, but lacks systematic and in-depth discussions. 

Comments on the Quality of English Language

Minor editing of English language required.

Round 2

Reviewer 2 Report

Comments and Suggestions for Authors

The author has seriously revised some of the suggestions. The revised content meets the requirements of this journal and is recommended to be accepted.

Comments on the Quality of English Language

The quality of English Language meets the requirements of this journal.

Author Response

Dear Reviewer!

We would like to express our deepest gratitude for your help in improving our manuscript and your positive feedback about it!

We really appreciate your work. Thank you!

Reviewer 3 Report

Comments and Suggestions for Authors

The authors have made some corrections in the revised manuscript. Some figures in Materials and test Methods were moved to Results and Discussion.

However, the relevant analysis and deep discussions have not been improved.

Comments on the Quality of English Language

Minor editing of English language required.
